# Study on the Recognition of Exercise Intensity and Fatigue on Runners Based on Subjective and Objective Information

**DOI:** 10.3390/healthcare7040150

**Published:** 2019-11-20

**Authors:** Guozhong Chai, Yinghao Wang, Jianfeng Wu, Hongchun Yang, Zhichuan Tang, Lekai Zhang

**Affiliations:** College of Mechanical Engineering, Zhejiang University of Technology, 310023 Hangzhou, China; chaigz@zjut.edu.cn (G.C.); wyhyk@126.com (Y.W.); yhc2106@zjut.edu.cn (H.Y.); ztang@zjut.edu.cn (Z.T.); zlkzhang@zju.edu.cn (L.Z.)

**Keywords:** running, fatigue, recognition, heart rate signal, EMG signal

## Abstract

A running exhaustion experiment was used to explore the correlations between the time-frequency domain indexes extracted from the surface electromyography (EMG) signals of targeted muscles, heart rate and exercise intensity, and subjective fatigue. The study made further inquiry into the feasibility of reflecting and evaluating the exercise intensity and fatigue effectively during running using physiological indexes, thus providing individualized guidance for running fitness. Twelve healthy men participated in a running exhaustion experiment with an incremental and constant load. The percentage of heart rate reserve (%HRR), mean power frequency (MPF) and root mean square (RMS) from surface EMG (sEMG) signals of the rectus femoris (RF), biceps femoris (BF), tibialis anterior muscle (TA), and the lateral head of gastrocnemius (GAL) were obtained in real-time. The data were processed and analyzed with the rating of perceived exertion (RPE) scale. The experimental results show that the MPF on all the muscles increased with time, but there was no significant correlation between MPF and RPE in both experiments. Additionally, there was no significant correlation between RMS and RPE of GAL and BF, but there was a negative correlation between RMS and RPE of RF. The correlation coefficient was lower in the constant load mode, with the value of only −0.301. The correlation between RMS and RPE of TA was opposite in both experiments. There was a significant linear correlation between %HRR and exercise intensity (r = 0.943). In the experiment, %HRR was significantly correlated with subjective exercise fatigue (r = 0.954). Based on the above results, the MPF and RMS indicators on the four targeted muscles could not conclusively identify fatigue of lower extremities during running. The %HRR could be used to identify exercise intensity and human fatigue during running and could be used as an indicator of recognizing fatigue and exercise intensity in runners.

## 1. Introduction

With the improvement of socio-economic status, fitness is being considered as a lifestyle [1]. Running is one of the most popular fitness regimens because it does not require equipment and can be practiced almost everywhere. Unfortunately, the annual incidence of injuries among runners is 40% to 50% [2]. Inappropriate exercise intensity during running is one of the leading causes of injury. Even worse, excessive exercise leads to local muscular injuries [3]. Through the monitoring and evaluation of exercise intensity and physical fatigue, scientific guidance during exercise is very useful and necessary to prevent injuries. To improve athletes’ performance and to cultivate talents in sports, the monitoring of training intensity and quantity in competitive sports requires precise and expensive scientific research instruments and one-to-one coaching with sports experts [4]. This approach is not suitable for amateur runners. For them, a set of convenient and feasible detection methods needs to be developed to guide their running.

Exercise intensity and fatigue can be evaluated using subjective self-perception, objective physiological, and mechanical indicators [5]. Self-perception is the self-evaluation [6] of intensity and fatigue through the rating of perceived exertion (RPE) [7]. The objective evaluation of exercise intensity is carried out by using direct indicators such as speed and force, and with the help of surface electromyography (sEMG) signals, percentage of maximum oxygen uptake (%VO_2_max), percentage of reserve heart rate (%HRR), blood lactic acid (BLA), and creatine kinase (CK). Other physiological and biochemical indicators can indirectly evaluate exercise intensity. The objective evaluation of exercise fatigue is obtained with blood urea (BU), hemoglobin (HGB), testosterone (T), sEMG signals, and heart rate [8]. sEMG and heart rate have the advantages of being non-invasive, real-time, and convenient, so they are widely used in the field of sports science and ergonomics. 

To study exercise-induced fatigue, especially localized muscle fatigue, sEMG technology uses the time and frequency domain indicators of the expanding signal. In the pedal exercise experiment with increasing load, Petrofsky et al. [9] found that there is a significant linear correlation between the amplitude of EMG root mean square (RMS) and muscle exercise load. The RMS increases continuously within a certain load range, while the median frequency (MF) decreases during the entire exercise. Wang et al. [10] proposed that the mean power frequency (MPF) of the frequency domain indicator decreased with fatigue, while the change of the MPF indicator could vary in the dynamic fatigue process. Tamaki et al. [11] proposed that the value of integrated EMG (iEMG) can better reflect the relationship between muscle fatigue and increase of exercise time. They also found that the frequency domain indicator (MPF) is more sensitive than the median frequency in measuring muscle fatigue.

During training, coaches monitor athlete training using the principle that the heart rate of athletes increases with the increase of intensity and fatigue. The percentage of heart rate reserve (%HRR) is included in the variable of resting heart rate of athletes. The difference in individual physical fitness can be better compared when monitoring and evaluating exercise intensity and exercise fatigue [12]. 

Previous studies have focused on heart rate indicators, sEMG indicators, exercise intensity, and fatigue. Most of these studies focus on static exercise or short-term isokinetic exercise, which strictly control the experimental conditions. Until now, research on the intensity and fatigue of long-term exercise for non-athlete individuals remain limited. Considering the lack of scientific fitness knowledge and guidance, this study used MPF, RMS of sEMG signal and %HRR, combined with the RPE scale in the experimental design. Increasing load exhaustive exercise is the most widely used experiment to evaluate the VO_2_ max. The %HRR has been highly correlated with %VO2max so it can replace %VO_2_ max in the evaluation and guidance of exercise intensity [13]. Therefore, an incremental load-running exhaustion experiment was designed to assess an appropriate personalized moderate running intensity using the %HRR interval and the mean personalized speed when the intensity is reached. In addition, a constant speed running exhaustion experiment was designed, in which the mean personalized speed, obtained in the incremental experiment, was used as the constant speed to explore the %HRR range at a moderate fatigue level. MPF and RMS were used to explore the localized muscle fatigue during the two experiments. The changing trend and correlation of each indicator in both experiments were observed, respectively. A suitable exercise intensity and fatigue degree for personalized running based on subjective and objective physiological information were further explored to provide real-time references for amateur runners. 

## 2. Experiment Design

### 2.1. Subjects

Twelve healthy male graduate students with no cardiovascular, cerebrovascular, respiratory, or musculoskeletal diseases were recruited. This study was aimed at amateur runners; therefore, the subjects have irregular fitness habits and no scientific guidance related to running. The basic morphological parameters of the subjects were as consistent as possible to reduce data differences caused by individual physical differences. The details are shown in Table 1.

### 2.2. Data Acquisition

#### 2.2.1. sEMG Signal Acquisition

The MP15 telemetry physiological recorder (American B20PAC Company) was used to collect the sEMG signal of lower limb muscles. According to the evaluation of exercise fatigue performance pointed out in previous studies [14], rectus femoris (RF), biceps femoris (BF), tibialis anterior muscle (TA), and lateral head of gastrocnemius (GAL) were selected as the target sites. These muscles are heavily engaged during running and provide different functions. The position of the muscle to be evaluated and the electrode location are shown in Table 2.

#### 2.2.2. Heart Rate Signal Acquisition

After connecting the electrode pieces on the transmission belt, the “Walker Pro” heart rate belt (Chinese Walker Outdoors Limited Company) was used to fix the electrode piece slightly to the left of the lower edge of the chest line. The sensor would automatically activate when the heartbeat was detected. The heart rate signal was transmitted to the ANT+ device via Bluetooth and recorded in real-time. 

#### 2.2.3. RPE Subjective Scale Value Collection 

A subjective scale was used to record the subjective psychological perception of intensity and fatigue during exercise. According to Carrie et al. [15], the relationship between subjective fatigue RPE value and exercise intensity and fitness effect was studied. The RPE scale is shown in Table 3. 

### 2.3. Experimental Process

Figure 1 showed experimental equipment and scene. The specific experimental process is shown in Figure 2.

#### 2.3.1. Collection of Resting Heart Rate

Resting heart rate refers to the number of beats per minute in a clear, inactive, quiet state. The subjects sat quietly for 5 minutes, and one minute after, the heart rate was collected [6]. The mean of the subjects’ resting heart rates were calculated.

#### 2.3.2. Pre-Running Exercise Guidance and Warm-Up

The subjects received guidance for using the treadmill, correct running posture, and breathing methods before running. These measures were implemented to avoid the influence of other factors besides physiological fatigue and exercise intensity. All the subjects warmed up properly before exercise to prevent strain or tendon tear during running [16].

#### 2.3.3. The Incremental Load Running Exhaustion Experiment and Data Acquisition

Each subject began to exercise at the speed of 6 km/h, the initial heart rate was recorded, and then the speed was increased by 0.5 km/h every 2 min. The heart rate values of the subjects were collected, and the subjective feelings of fatigue and exercise intensity were asked every minute. The observer recorded the both kinds of RPE values currently. The target muscle of the subjects was connected to the sEMG collector to gather and transmit the sEMG data to the computer. The experiment lasted until the subjects could not continue to run.

#### 2.3.4. The Constant Load Running Exhaustion Experiment and Data Acquisition

Previous studies have proposed that when the exercise intensity RPE reaches the range of 13 to 15, the exercise intensity is moderate, and the best fitness effect can be obtained by maintaining the exercise intensity in this RPE interval [17]. Therefore, in each subject, the speed of exercise intensity RPE in the range of 13 to 15 was recorded to obtain the mean as a constant load of exhaustion for the personalized running speed. After the incremental load exhaustion experiment, the subjects needed time to rest and recover. During the recuperation time, the subjects were requested for feedback about their physical fatigue every day. After six days, it was determined that all the subjects eliminated their fatigue. At this point the constant load experiment was started. The subjective fatigue RPE value of the subjects was recorded every other 2 min until they were exhausted. sEMG signal and heart rate were collected during the entire process.

#### 2.3.5. Data Processing and Analysis

Due to the different completion time of each experiment, the data during exercise were standardized, and the individual exhaustion time was 100%T.

HR max was obtained following a minimum error calculation method of HR max = 208 − 0.7 × age, which was used to calculate the maximum heart rate of ordinary athletes without professional training [18]. Combined with the resting heart rate measured before exercise and the real-time heart rate at each time point, %HRR value was obtained with the following formula [19]: %*HRR* = (*real-time heart rate during exercise-resting heart rate*)/(*HR max-resting heart rate*).(1)

During the experiment, %HRR was extracted at each time point to compare with other data, and the correlation analysis was performed. Simultaneously, the frequency domain indicator (MPF) and time-domain indicator (RMS) of sEMG signal were extracted. The formula for calculating the frequency domain indicator is as follows [20]:
(2)$$MPF=\frac{\int_{0}^{\infty }f\cdot P\left(f\right)df}{\int_{0}^{\infty }P\left(f\right)df}$$

The calculated formula of the time-domain indicator is as follows [21]:(3)$$RMS=\sqrt{\frac{1}{N}\int_{t}^{t+T}EMG^{2}\left(t\right)\cdot dt}$$
*t* represents the sampling time of the experimental data.

The change characteristics of %HRR, MPF, RMS, and subjective fatigue RPE of the four targeted muscles in both experiments were analyzed, respectively. The correlation between the collected data of indicators and regression analysis was carried out using SPSS version 19.0 (IBM, Corporation, Armonk, USA).

## 3. Results

### 3.1. The Incremental Load Exhaustion Experiment

Among the 12 subjects, the longest exercise time was 24 min while the shortest exercise time was 15 minutes. The highest speed was 12 km/h, and the lowest speed was 9.5 km/h.

#### 3.1.1. Change Characteristics of %HRR in the Incremental Load Exhaustion Experiment

Figure 3 shows that %HRR increased gradually along with the increase of time and exercise intensity. The growth rate of %HRR was fast at the beginning and slowed down in the progress. Finally, the maximum heart rate was (100.12 ± 5.23) %HR max. The correlation analysis showed a significant linear correlation between %HRR and standardized time T% (r = 0.943).

In the experiment, exercise intensity RPE increased with the increase of time. The correlation analysis between %HRR and exercise intensity RPE showed a very significant correlation (r = 0.954, *p* < 0.001). The mean %HRR of appropriate intensity RPE in the range of 13 to 15 was (74.59 ± 7.37) to (82.87 ± 8.64) %HR max.

#### 3.1.2. Change Characteristics of MPF in the Incremental Load Exhaustion Experiment

As shown in Figure 4, during the incremental load experiment, the MPF of all the targeted muscles fluctuated to a certain extent and showed a slight upward trend.

As shown in Table 4, in the incremental load experiment, except for MPF of TA and GAL, MPF of RF and BF were correlated with the fatigue RPE of the subjects (*p* < 0.05). However, the correlation coefficients were low.

#### 3.1.3. Change Characteristics of RMS in the Incremental Load Exhaustion Experiment

As shown in Figure 5, with the increase of time, the changing trend of root mean square (RMS) in each muscle was different. RMS of RF showed a downward trend, RMS of BF and TA fluctuated, but the overall trend was stable. RMS of GAL showed a slight upward trend. Further correlation analysis between RPE and RMS of the targeted muscle is shown in Table 5. There was no significant correlation between RMS of BF, TA, GAL, and subjective fatigue RPE. Then RMS of RF and RPE showed a moderate negative correlation, but not statistically significant.

### 3.2. The Constant Load Exhaustion Experiment

In the incremental load experiment, when the subjects reached the “appropriate intensity” (RPE value of 13 to 15), the speed mean values of the 12 subjects were calculated respectively. Each speed mean value was used as a fixed speed in the constant load experiment. In the constant load experiment, the longest exercise time of 12 subjects was 68 minutes, while the shortest exercise time was 36 minutes.

#### 3.2.1. Change Characteristics of %HRR in the Constant Load Exhaustive Experiment

As shown in Figure 6, in the constant load exhaustion experiment, %HRR increased rapidly in the range of 0 to 30%T, and slowly in the range of 30 to 80%T until the subjects were exhausted. The heart rate did not reach 100%HRR in this experiment. The standardized time T% was significantly correlated with %HRR (r = 0.842).

With the increase in exercise time and exercise amount, there was a significant positive linear correlation between subjective fatigue RPE and T% (r = 0.973). The %HRR corresponding to subjective fatigue RPE is shown in Figure 7, and there was a very significant linear correlation between subjective fatigue RPE and %HRR (r = 0.910). When running to tiredness, that is moderate fatigue, the mean value of %HRR of subjective fatigue RPE in the range of 15 to 16 was (76.03 ± 8.35) to (84.33 ± 7.56) % HR max.

#### 3.2.2. Change Characteristics of MPF in the Constant Load Exhaustive Experiment

As shown in Figure 8, with increasing time, the MPF of the four targeted muscles showed a certain upward trend. Among them, the MPF of BF and RF increased steadily, while MPF of TA and GAL fluctuated slightly. However, the overall MPF increased compared with the initial time.

As shown in Table 6, among the four targeted muscles, only MPF of RF and subjective fatigue RPE showed a weak correlation, and the correlation coefficient was low. However, there was no significant correlation between MPF of BF, GAL, and subjective fatigue RPE, and there was almost no correlation between MPF of TA and subjective fatigue RPE.

#### 3.2.3. Change Characteristics of RMS in the Constant Load Exhaustive Experiment

As shown in Figure 9, the RMS of the four targeted muscles showed a significant downward trend. Among them, the RMS curves of RF and TA were similar, and the overall trend was relatively stable. The RMS of BF and GALs was higher than that of RF and TA and fluctuated slightly.

As shown in Table 7, there was a weak negative correlation between RMS of RF and RPE with a low correlation coefficient. There was a moderate negative correlation between RMS of TA and RPE, and there was no significant correlation between RMS mean of BF, GAL, and RPE.

## 4. Discussion

### 4.1. Analysis and Discussion on the Experimental Results of sEMG Signal

#### 4.1.1. Analysis and Discussion on the Results of the Frequency Domain Indicator (MPF) of the sEMG Signal

The results of both experiments showed that with exercise progression, MPF of the four targeted muscles fluctuated differently, but showed a certain upward trend. MPF reflects the changes in muscle sEMG signals in the frequency dimension during exercise. In general, with increasing exercise muscle fatigue, MPF and MF, which reflect the characteristics of the Fourier spectrum curve of sEMG signals, decrease accordingly, resulting in varying degrees of spectrum curve left shift [22,23]. MPF change is thought to be related to the hydrogen ion produced by muscles during fatigue. The concentration of hydrogen ion in muscle usually increases with exercise fatigue, which slows down the action potential conduction rate and decreases the frequency component of sEMG, shown as the decrease of MPF [24]. However, some scholars have found that MPF shows other trends during exercise. Yang et al. evaluated 12 healthy men with squatting exercise at maximum load and found that the concentric and isometric contraction of the lateral thigh muscle increased at the beginning and then decreased with the decrease of exercise ability during the period of centripetal and isometric contraction of the lateral thigh muscle [25]. Ament et al. designed an incremental load experiment on a treadmill and found that MPF of the gastrocnemius and soleus muscle experienced no significant change during exhaustive exercise from low to high [26]. 

Possible reasons for the upward trend analysis of MPF are as follows. The exercise intensity in the incremental load experiment increased with time, resulting in a sharp increase in cardiopulmonary expenditure, and the subjective feeling reached the fatigue limit in a short time. Because the exercise time was short, the leg muscle did not reach the fatigue state, so the MPF did not show the downward trend. In the incremental load experiment, the treadmill speed was increasing, and the subjects needed to increase the step frequency in order to keep up with speed, which might also be one of the reasons for the upward trend of MPF. In the constant load experiment, the rise of MPF might show the effect of the central nervous system (CNS) on the sEMG signal during extended dynamic exercise. Previous studies have shown that the frequency domain indicator of sEMG is affected by peripheral hydrogen ions, as well as CNC impulse emission frequency [27,28]. Zhang found a phenomenon of central co-drive in the process of muscle exercise fatigue [29]. During running, the discharge frequency of CNS might increase, and the targeted muscle was affected by co-driving so that the frequency domain indicator (MPF) did not show the descending mode of static fatigue.

In the incremental load experiment, there was a weak correlation between MPF of RF, BF, and subjective fatigue RPE, but there was no correlation between MPF of TA, GAL and fatigue RPE. In the constant load experiment, only MPF of RF and fatigue RPE showed a weak correlation with a low correlation coefficient. In both experiments, MPF of all the muscles showed no fatigue characteristics during exercise (the decrease of MPF). Therefore, the MPF indicator of the four targeted muscles could not reflect and evaluate running fatigue.

#### 4.1.2. Analysis and Discussion of the Results of the Time-Domain Indicator (RMS) of the sEMG Signal

The time-domain indicator (RMS) reflected the change in amplitude of the targeted muscle sEMG in the time dimension, that is, the effective discharge value. The results showed that in the incremental load experiment, the changing trend of RMS in the four muscles was different, and the mean of RMS of GAL, and TA showed an upward trend. RMS of BF showed no significant difference before and after exercise, while RMS of RF decreased significantly. In the constant load experiment, RMS of the four targeted muscles showed a certain downward trend.

Previous studies have pointed out that the size of RMS is determined by the strength and discharge of muscles during exercise, and different modes of exercise will lead to different trends of RMS. Under the same exercise mode, when athletes exercise at maximum intensity, the discharge of muscle may decrease with fatigue. When athletes exercise at sub-high intensity, increased fatigue will stimulate muscles to increase electricity discharge to compensate for decreased muscle strength, resulting in increased RMS [30,31]. However, when the exercise intensity is low, there is a great difference in RMS among different subjects with muscle fatigue. Some subjects show an increasing linear change, other subjects show a decreasing change, and some other subjects do not even produce any significant change [10].

Considering the large variability of RMS changes in the experiment, the results showed that the correlation between RMS and fatigue RPE was low and mostly not significant. This study concluded that RMS of the four targeted muscle was not suitable for common daily exercise to identify the fatigue state.

### 4.2. Analysis and Discussion of the Experimental Results of %HRR

#### 4.2.1. Analysis of the Relationship between %HRR and Exercise Intensity

Based on the results of incremental load experiment, the correlation analysis showed a high linear correlation between standardized time T% and %HRR (r = 0.943, *p* < 0.001), but the change of exercise intensity and T% was constant. Therefore, the experimental results showed that %HRR could identify the exercise intensity in incremental load running and could be used as a useful indicator to identify and evaluate exercise intensity in daily running. These results are consistent with domestic and foreign studies that mention a positive correlation between heart rate and running speed.

The mean value of %HRR was (74.59 ± 7.37) to (82.87 ± 8.64) %HR max in the RPE range of moderate intensity in the incremental load experiment, while the %HRR was (77.03 ± 8.35) to (85.33 ± 7.56) %HR max in the constant moderate fatigue state. Therefore, subjects still need to exercise for a period in order to achieve a moderate fatigue state when running at a moderate intensity.

Generally, HR max can only be directly measured by increasing its intensity to the limit [32]; the overall intensity exercise cannot directly reach the HR max. At the end of the incremental experiment, the subjects reached (100.12 ± 5.23) %HR max, while at the end of the constant experiment, the subjects significantly failed to reach HR max.

#### 4.2.2. Analysis of the Relationship between %HRR and Exercise Fatigue

When detecting exercise fatigue in the constant load experiment, the correlation coefficient between %HRR and subjective fatigue RPE was 0.910, *p* < 0.01, indicating that %HRR indicator could be used to identify the fatigue state of athletes during running. The value of %HRR in the constant experiment was (77.03 ± 8.35) to (85.33 ± 7.56) %HR max, which was following the moderate fatigue range of the actual running exercise. The results verified the aerobic training intensity proposed by the American Institute of Physical Education that mentions the targeted training should be equivalent to 65% to 90% of HR max to improve an athletes level [6].

Combined with the analysis of the relationship between %HRR and exercise intensity, and between %HRR and exercise fatigue, %HRR indicator could reflect the exercise intensity and fatigue state of athletes during running. Furthermore, %HRR could also be used as a personalized indicator to identify and evaluate how people run. Therefore, it can be used in real-time running training.

## 5. Conclusions

First, this study showed that MPF and RMS indicators of sEMG signal during running could not conclusively identify the fatigue level of the targeted muscle. These indicators were not suitable for use as a reference for personalized running assessment.

Second, %HRR could effectively identify and evaluate the exercise intensity and fatigue of runners to a certain extent. It is suggested that the personalized running speed of different individuals with appropriate intensity could be measured in the form of incremental running speed according to the mean interval of %HRR of appropriate intensity 75%–80% HRmax, and the mean range of %HRR could be obtained from personalized running speed exercise in moderate fatigue, with a recommended range of 77%–85% HRmax. Individuals running at a personalized speed of appropriate intensity to moderate fatigue %HRR will achieve better fitness results and prevent excessive fatigue.

Lastly, the physiological indicators selected in this study are limited. In futures studies, wearable detecting techniques such as infrared sensing and respiratory telemetry can be implemented to identify the running status of runners, which could enable personalized scientific guidance for ordinary runners.

## Figures and Tables

**Figure 1 healthcare-07-00150-f001:**
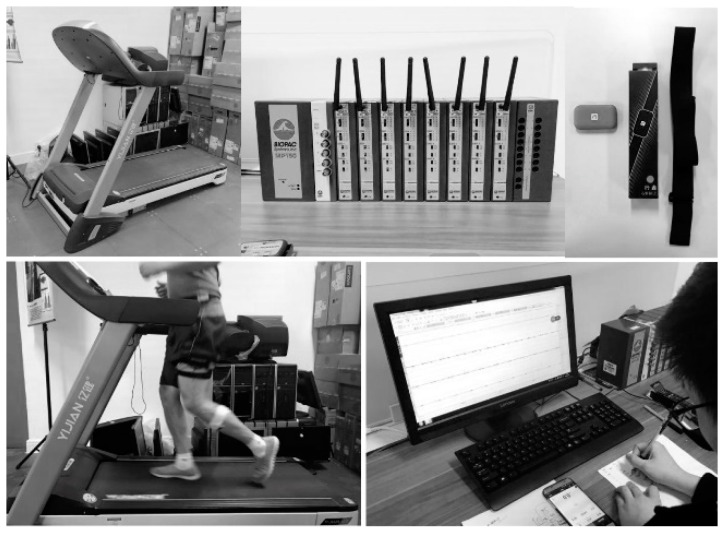
Experimental equipment and recording scene.

**Figure 2 healthcare-07-00150-f002:**
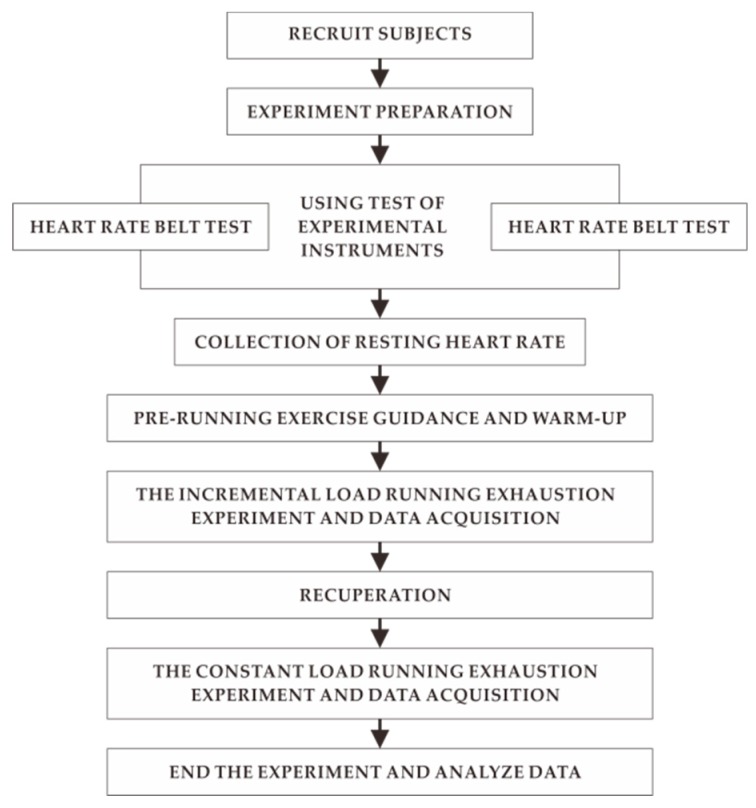
Experimental progress.

**Figure 3 healthcare-07-00150-f003:**
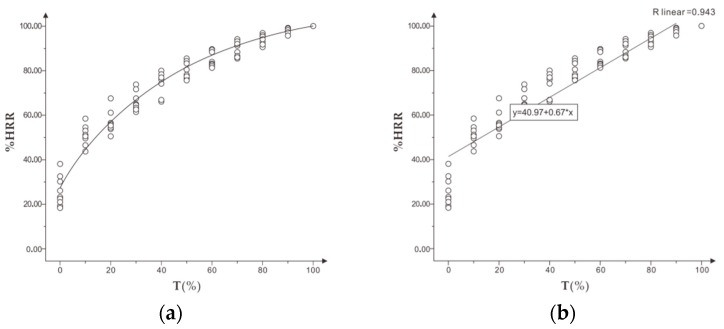
(**a**) shows the trend of %HRR in standard time and (**b**) shows the correlation between %HRR and T% in the incremental load experiment.

**Figure 4 healthcare-07-00150-f004:**
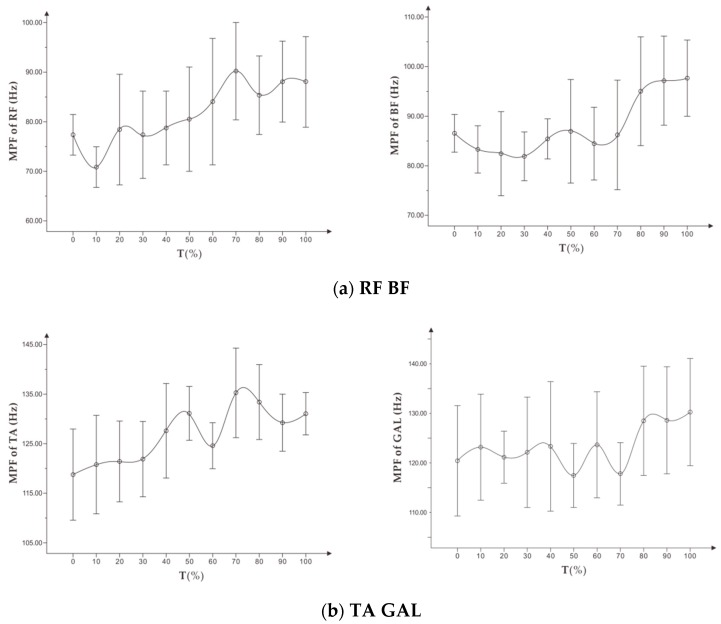
(**a**) shows the trend of MPF ($\bar{x}$ ± s) on RF, BF, and (**b**) shows the trend of MPF ($\bar{x}$ ± s) on TA, GAL in the incremental load experiment.

**Figure 5 healthcare-07-00150-f005:**
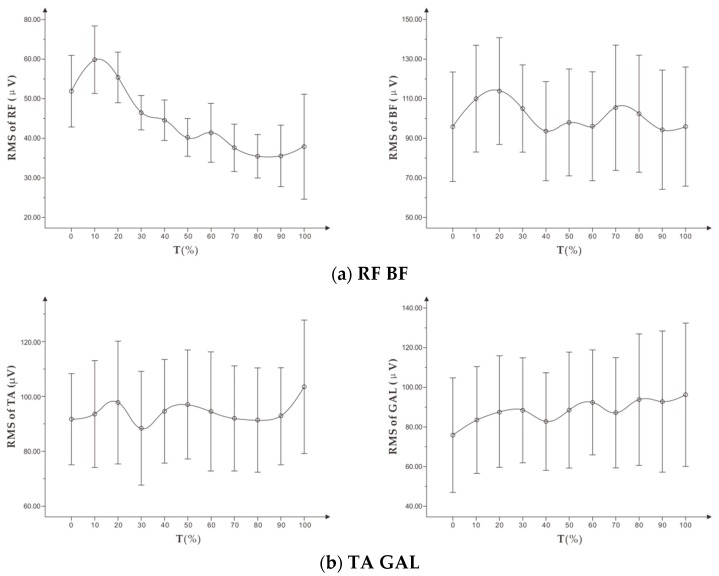
(**a**) shows the trend of RMS ($\bar{x}$ ± s) on RF, BF, and (**b**) shows the trend of RMS ($\bar{x}$ ± s) on TA, GAL in the incremental load experiment.

**Figure 6 healthcare-07-00150-f006:**
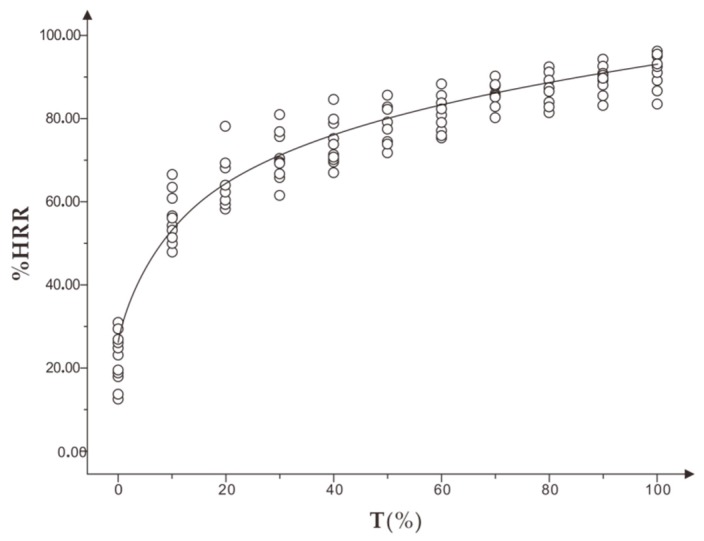
The trend of percentage heart rate reserve (%HRR) in standardized time in the constant load experiment.

**Figure 7 healthcare-07-00150-f007:**
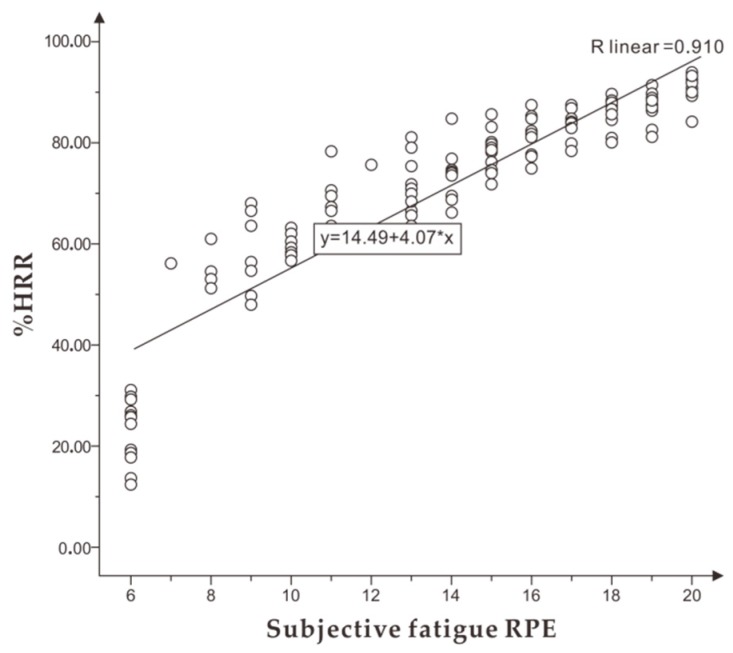
The correlation between %HRR and subjective fatigue RPE in the constant load experiment.

**Figure 8 healthcare-07-00150-f008:**
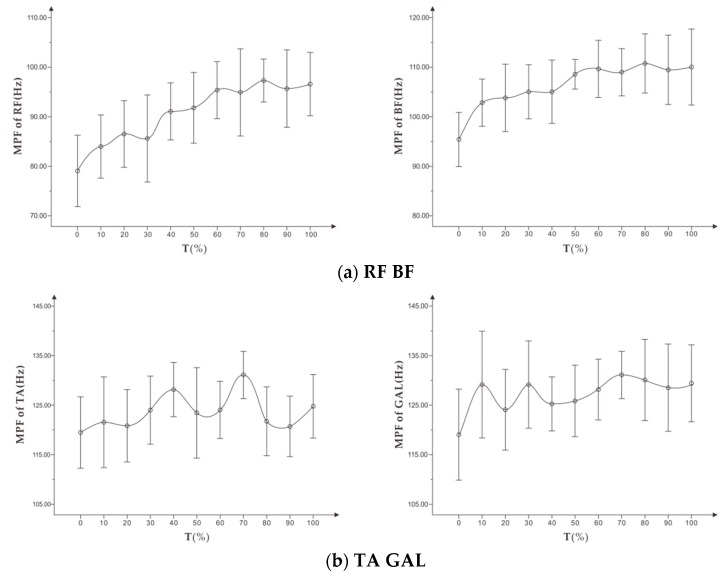
(**a**) shows the trend of MPF ($\bar{x}$ ± s) on RF, BF, and (**b**) shows the trend of MPF ($\bar{x}$ ± s) on TA, GAL in standardized time in the constant load experiment.

**Figure 9 healthcare-07-00150-f009:**
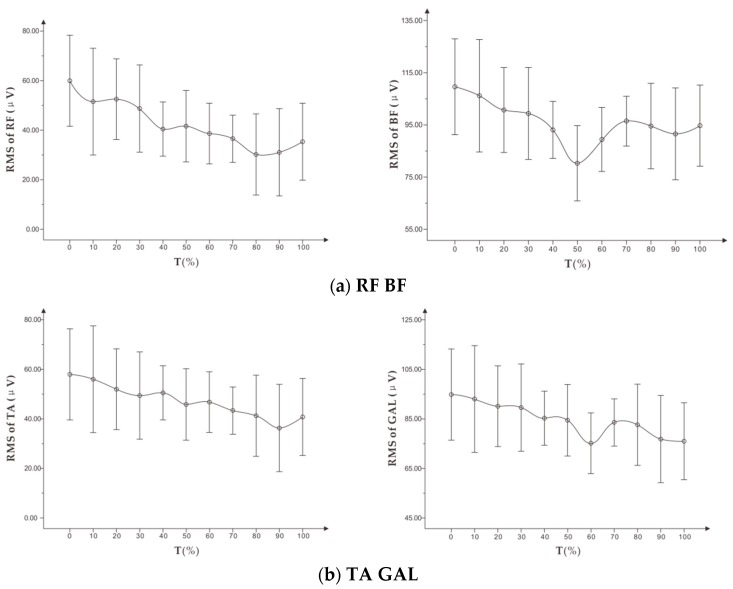
(**a**) shows the trend of RMS ($\bar{x}$ ± s) on RF, BF, and (**b**) shows the trend of RMS ($\bar{x}$ ± s) on TA, GAL in standardized time in the constant load experiment.

**Table 1 healthcare-07-00150-t001:** Subjects’ basic information ($\bar{x}$ ± s).

Number of Subjects	Age (Years)	Height (cm)	Weight (kg)	Resting Heart Rate (Beats/min)
12	25.25 ± 1.93	173.83 ± 1.75	67.25 ± 3.91	76.17 ± 8.28

**Table 2 healthcare-07-00150-t002:** Muscle and the electrode location.

Name of the Target Muscle	RF	BF	TA	GAL
Electrode location	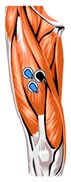	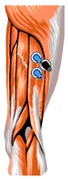	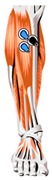	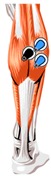

**Table 3 healthcare-07-00150-t003:** Subjective perceived exertion scale (RPE).

Evaluation Grade	Subjective Exercise Intensity	Subjective Exercise Fatigue
6	Almost no exercise intensity	Not hard at all
7		Extremely relaxed
8		
9	Very low exercise intensity	Very relaxed
10		
11	Low exercise intensity	Relaxed
12		
13	Appropriate exercise intensity	A little tired
14		
15		Tired
16	High exercise intensity	
17	Secondary maximum intensity	Very tired
18		
19	Maximum intensity	Extremely tired
20		Try the best

**Table 4 healthcare-07-00150-t004:** The correlation between MPF and subjective fatigue (RPE) of each targeted muscle in the incremental load experiment.

	MPF of RF	MPF of BF	MPF of TA	MPF of GAL
Spearman’s Rho	0.368 *	0.364 *	0.179	0.336
Sig.(2-tailed)	0.047	0.029	0.524	0.069

* 0.05 level (2-tailed) significant correlation.

**Table 5 healthcare-07-00150-t005:** The correlation between RMS and subjective fatigue RPE of each targeted muscle in the incremental load experiment.

	RMS of RF	RMS of BF	RMS of TA	RMS of GAL
Spearman’s Rho	−0.514 *	−0.050	0.132	−0.029
Sig.(2-tailed)	0.050	0.860	0.639	0.919

* 0.05 level (2-tailed) significant correlation.

**Table 6 healthcare-07-00150-t006:** The correlation between MPF and subjective fatigue RPE of each targeted muscle in the constant load experiment.

	MPF of RF	MPF of BF	MPF of TA	MPF of GAL
Spearman’s Rho	0.221 *	−0.043	0.018	0.093
Sig.(2-tailed)	0.039	0.879	0.950	0.742

* 0.05 level (2-tailed) significant correlation.

**Table 7 healthcare-07-00150-t007:** The correlation between RMS and subjective fatigue RPE of each targeted muscle in the constant load experiment.

	RMS of RF	RMS of BF	RMS of TA	RMS of GAL
Spearman’s Rho	−0.279	−0.096	−0.432 *	−0.107
Sig.(2-tailed)	0.315	0.372	0.031	0.704

* 0.05 level (2-tailed) significant correlation.

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
