# Peer review of "Study on the Recognition of Exercise Intensity and Fatigue on Runners Based on Subjective and Objective Information"

_healthcare, 2019, doi:10.3390/healthcare7040150_

Round 1

Reviewer 1 Report

I have the following comments on “Study on the Recognition of Exercise Intensity and Fatigue on Runners Based on Subjective and Objective Information”. Chai et al calibrated a running exhausting experiment in 12 healthy amateur runners with subjective exhaustion information and correlated it with data of heart rate and electromyogram signals. They found that “heart rate reserve”, exercise intensity and subjective exercise fatigue strongly correlated with each other. However, they found no parameters of electromyogram signals to be correlated with exercise intensity.

This study has major issues, especially concerning the methodology and reporting. Therefore, I would not consider it for publication in its present form.

It is unclear to me how the study itself was carried out. It is reported that “After the incremental load exhaustion experiment, the subjects returned every day. After six days, […]”. What does that mean? Which data was used? Just data of the last day? Please provide a detailed flow chart to present every step in the conducting the study and analysis. There was no correlation between the EMG signals and exercise intensity. This may be explained by the following factors. First, the investigated parameters may be calculated wrongly. How did you come up with the formulas in page 5? There are no references for RMS and MPF given, and %HRR is not defined in reference [19]. Please give exact citations. Please define “basal heart rate” and “quiet heart rate” and the difference. Please provide a figure with an example signal. Second, sEMG signals may be correlated with exercise intensity, but not in a linear way. Why do you use the pearson coefficient? Please use Spearman’s rho as correlations may be non-linear. Third, I cannot understand why all sEMG signals were NOT investigated with subjective perception with fatigue (as implied by the title), but with exercise intensity. What are the authors’ thoughts about this issue? There are major spelling and reporting errors. Just do give examples: sEMG vs. SEMG; Desing vs Design. Some references are cited wrongly, for example [19] (first and last name mixed up – please check the remaining references). “The growth rate of %HRR was fast and slow […]”? Minor issues: Please defined the “walker” heart rate belt (type? Company?) Please improve all plots (remove Chinese words), add Spearman’s rho, p values and add plots of significant correlations. What does the following sentence mean: “The basal heart rate did not reach 100% HRR”? P values should always have the same number of digits (Table 5). Please give definitions to all abbreviations directly in the description of the tables/figures. Explain the “*”.

Author Response

Dear reviewer:

Thank you very much for your valuable comments about our paper submitted to the healthcare-611816.

We thank you for your encouraging and professional review of our previous  manuscript. We have carefully considered the comments and have revised the manuscript accordingly. The relevant problems had been revised in the original manuscript according to your comments. We also responded point by point to each comment as listed below.

It is reported that “After the incremental load exhaustion experiment, the subjects returned every day. After six days, […]”. What does that mean? Which data was used? Just data of the last day? Please provide a detailed flow chart to present every step in the conducting the study and analysis.?

Response: We are sorry. It's a mistake in translation that leads to misunderstanding,especially the word “return”. What we want to express is “After the incremental load exhaustion experiment, the subjects needed time to rest and recover. During the recuperation time, the subjects were requested for feedback about their physical fatigue every day. After six days, it was determined that all the subjects eliminated their fatigue. And then the constant load experiment was started." This problem has been revised in original manuscript which in line 140-143. We used the data which was collected during our experiment. We had not collect any data during the recuperation time. In order to make the experimental process clearer,we drew a detailed flow figure “Figure 2” in line 118.

There was no correlation between the EMG signals and exercise intensity. This may be explained by the following factors. First, the investigated parameters may be calculated wrongly. How did you come up with the formulas in page 5? There are no references for RMS and MPF given, and %HRR is not defined in reference [19]. Please give exact citations. Please define “basal heart rate” and “quiet heart rate” and the difference.Please provide a figure with an example signal. Second, sEMG signals may be correlated with exercise intensity, but not in a linear way. Why do you use the pearson coefficient? Please use Spearman’s rho as correlations may be non-linear. Third, I cannot understand why all sEMG signals were NOT investigated with subjective perception with fatigue (as implied by the title), but with exercise intensity. What are the authors’ thoughts about this issue?

Response: Please allow us to answer your “Third suggestion” first. We are sorry that we had not described the experimental progress and the results accurately in our previous manuscript. Actually, previous studies had proved that exercise intensity is highly correlated with %HRR. Coaches usually use %HRR to define athletes' exercise intensity. And it’s mentioned on page 144 of the ACSMs guidelines for exercise testing and prescription (tenth edition). So, we explored the %HRR numerical range of suitable exercise intensity in incremental load exhaustion experiment. sEMG is highly correlated with exercise intensity especially in static and isometric exercise or short-term isokinetic exercise. However, the trend of sEMG is variable during the long-term running exercise. Therefore we did not analyze the correlation between the sEMG signals and exercise intensity. sEMG is usually used to explore the localized muscle fatigue in the field of human kinematics and sports medicine. Therefore we analyzed the correlations between sEMG signals and subjective fatigue to explore the possibility of using EMG signal to reflect localized muscle fatigue in this paper. There are two kinds of RPE in our paper. One is “exercise intensity RPE” and the other one is “exercise fatigue RPE”. The results in Table3-6 shows the correlation between sEMG signals and subjective fatigue RPE. We are so sorry that you might misunderstand it because of our improper description. We have improved all the description in the “1. Introduction, line33-34, line51, line79-80” , “Chapter2.3.3., line 132”, “Chapter2.3.4., line138, line 143-144”, “3. Results”.

Then let’s answer your “First suggestion”. The formulas in Paper5-6 are certainly right. In the new revised manuscript, we had given the exact citations of formulas and %HRR in reference [19],[20],[21], according to your professional advice. Both the “basal heart rate” and “quiet heart rate” represent the “resting heart rate” in this paper and we gave the define in line 120. We unified the “resting heart rate” which in Table1, line 119, line 122, line 151 and line 153 to eliminate the ambiguity.

Finally, let’s answer your “Second suggestion”. Your suggestion about using Spearman’s rho as correlations is very valuable. All the authors agreed with you. There is no correlation between sEMG signals and subjective fatigue in the previous manuscript. We’d like to find some interesting results. According to your suggestion, we recalculated the correlations using Spearman’s rho based on the source data of experiment. The new results is displayed in Table3-6. Every numerical value has been changed. But unfortunately, the results of new calculation shows that the correlations between sEMG signals and fatigue RPE essentially unchanged. Then we revised the manuscript based on the new results.

3. There are major spelling and reporting errors. Just do give examples: sEMG vs. SEMG; Desing vs Design.

Response:Thank you for your careful review! We have carefully examined all the words and revised the errors in line 84, line 144.

4. Some references are cited wrongly, for example [19] (first and last name mixed up – please check the remaining references).

Response:Thank you for your careful review! We have carefully examined all the references and revised the errors.

5. “The growth rate of %HRR was fast and slow […]”?

Response:It’s a translation error. We have revised the errors in line171-172. “The growth rate of %HRR was fast at the beginning and slowed down in the progress.

6. Minor issues: Please defined the “walker” heart rate belt (type? Company?)

Response:Thank you for your suggestion! We have added the detailed information of heart rate belt in line102-103. Walker Pro” heart rate belt (Chinese Walker Outdoors Limited Company)”.

7. Please improve all plots (remove Chinese words), add Spearman’s rho, p values and add plots of significant correlations.

Response:We have improved all plots and updated all the figures “Figure3-9” which contained Chinese words. And also, we have add Spearman’s rho, p values in Table3-6 and revised the plots about the new results.

8. What does the following sentence mean: “The basal heart rate did not reach 100% HRR”?

Response:It’s a translation error. We have revised it in line221-222. The heart rate did not reach 100%HRR in this experiment”.

P values should always have the same number of digits (Table 5).Please give definitions to all abbreviations directly in the description of the tables/figures. Explain the “*”.

Response:We have revised the format errors in all tables. “* 0.05 level(2-tailed), significant correlation. This sentence has been added after Table3-6.

Overall, thank you very much for your valuable comments, which made this paper better. Hope these will make it more acceptable for publication. If you have any question about this paper, please do not hesitate to let me know. Thank you!

Sincerely yours,

Jianfeng Wu

Reviewer 2 Report

This study conducted a comprehensive investigation of the relationships among that the features of heart rate, EMG and subjective evaluations.
The authors revealed that %HRR could be an indicator to get the exercise intensity and human fatigue.
The findings of this paper can contribute to many kinds of research focusing on fatigue estimation, EMG analysis and Heart rate analysis.

The followings are the comments.

In Table 5, 6, please align the row between the name and values.

Line 135, “SEMG” is “sEMG”?

Figure 2, Figure 3, Figure 4, Figure 5, Figure 7, Figure 8, please write the label of x-axis and y-axis in English.

Author Response

Dear reviewer:

Thank you very much for your valuable comments about our paper submitted to the healthcare-611816.

We thank you for your encouraging and careful review of our previous  manuscript. We have carefully examined the errors and have revised the manuscript accordingly. The relevant problems had been revised in the original manuscript according to your comments. We also responded point by point to each comment as listed below.

In Table 5, 6, please align the row between the name and values.

Response: This is a format error and we have revised it that all words is aligned in Table1-6.

Line 135, “SEMG” is “sEMG”?

Response: Yes, it’s an error. It should be “sEMG” and we have revised in line144.

3. Figure 2, Figure 3, Figure 4, Figure 5, Figure 7, Figure 8, please write the label of x-axis and y-axis in English.

Response:We have added a figure “Figure 2” which displays the detailed flow. And then, we updated all the figures “Figure3-9” in which the label of x-axis and y-axis is written in English.

In order to make this paper better, we have carefully examined the other errors and have improved all the plots in the new revised manuscript. What’s more, we have added more details and improve the descriptions in the paper to clear up any misunderstandings.

We believe that the additional changes we have made in response to your comments have made this a significantly stronger manuscript.

Overall, thank you very much for your valuable comments, which made this paper better. Hope these will make it more acceptable for publication. If you have any question about this paper, please do not hesitate to let me know. Thank you!

Sincerely yours,

Jianfeng Wu

Round 2

Reviewer 1 Report

All previous comments have adequately been addressed.